# Digital PCR-Based Characterization of a *g10evo-epsps* Gene-Specific Matrix Reference Material for Its Food and Feed Detection

**DOI:** 10.3390/foods11131888

**Published:** 2022-06-25

**Authors:** Xiaoyun Chen, Huiru Yu, Pengfei Wang, Cheng Peng, Xiaofu Wang, Xiaoli Xu, Junfeng Xu, Jingang Liang, Liang Li

**Affiliations:** 1State Key Laboratory for Managing Biotic and Chemical Threats to the Quality and Safety of Agro-Products, Zhejiang Academy of Agricultural Sciences, Hangzhou 310021, China; xiaoyunchen_2016@163.com (X.C.); pc_phm@163.com (C.P.); yywxf1981@163.com (X.W.); xuxiaoli@zju.edu.cn (X.X.); njjfxu@163.com (J.X.); 2College of Chemistry and Life Sciences, Zhejiang Normal University, Jinhua 321001, China; yhr_yee@163.com; 3Institute of Insect Sciences, College of Agriculture and Biotechnology, Zhejiang University, Hangzhou 310058, China; 0620669@zju.edu.cn; 4Development Center of Science and Technology, Ministry of Agriculture and Rural Affairs, Beijing 100176, China; 5Institute of Quality Standard and Testing Technology for Agro-Products, Chinese Academy of Agricultural Sciences, Beijing 100081, China

**Keywords:** certified reference material, ZUTS-33 soybean, *g10evo-epsps*, digital PCR, glyphosate resistance

## Abstract

*g10evo-epsps* is a novel glyphosate herbicide-resistant gene that has been transferred to various crops such as soybean, corn, cotton, and rice. Here, we developed a gene-specific digital Polymerase Chain Reaction (dPCR) detection method for absolute quantitative analysis of *g10evo-epsps*, and characterized *g10evo-epsps* certified reference materials (CRM) using ZUTS-33 soybean powder as the candidate material. Stability tests of matrix CRMs demonstrate that these CRMs can be stored stably for 6 months and transported for 10 days at room temperature and withstand summer high temperatures (below 60 °C). CRM characterization is based on the copy number ratio of *g10evo-epsps* to *lectin*. Eight qualified laboratories independently validated the CRM using dPCR method, with a measurement of 0.98 (copy/copy) and an extended uncertainty of 0.08 (copy/copy). The *g10evo-epsps* matrix CRM described here may be used for qualitative and quantitative testing, method evaluation, laboratory quality control, and other related fields.

## 1. Introduction

Since the approval of genetically modified soybeans for commercial cultivation in 1996, Genetically modified organisms (GMOs) have transformed the industrialization of agriculture on a global scale; the number of GMO crops and related products grown commercially worldwide has developed rapidly [1]. However, the safety of GMO crops has been of widespread community concern. In order to protect consumers’ right to knowledge and choice, some countries require that genetically modified ingredients must be indicated [2]. GMO labelling requirements vary widely across nations; therefore, in order to resolve trade disputes and other issues, accurate and precise qualitative and quantitative testing of GMO products is essential [3,4,5].

Insect resistance and herbicide tolerance are two important genetically modified traits [6]. For example, glyphosate tolerance, the most widely planted GMO, is derived from bacterial EPSP synthase variants from *Agrobacterium tumefaciens* sp. CP4 [7]. Likewise, researchers are committed to developing novel resistant genes with their own intellectual property rights. *g10evo-epsps* is a highly glyphosate-resistant gene discovered by Zhejiang University which has been transformed into a variety of crops and has demonstrated strong resistance to glyphosate. These *g10evo-epsps* crops are in the process of safety certification and commercialization (Table 1). Corresponding testing methods and reference materials for this novel GMO are continuing to be developed.

Certified reference materials (CRMs) are certified reference materials (RMs) [8] (ISO Guide 30) characterized using a metrologically validated procedure for one or more specific properties, which can be used for calibration of a measurement system, assessment of a measurement procedure of other materials, and quality control. Genetically modified RMs consist of target gene DNA; this is the “characteristic value.” With the development of nucleic acid measurement technology, characteristic values are quantified with qPCR and digital PCR (dPCR) technology. dPCR methods, such as microfluidic chamber-based or droplet-based dPCR, are capable of absolute quantification of nucleic acids without the use of standard curves [9], thus these technologies have the potential to become a primary method (JCGM 200:2012) [10] for estimating DNA copy number [11,12].

The CRMs of genetically modified plants are mainly developed in three types: matrices (seed powder), genomic DNA, and plasmid DNA [13,14]. Each has advantages and disadvantages under the conditions of development, valuation, storage and transportation. The preparation of matrix RMs is complex [15,16,17], but powder is stable in storage and easy to transport. Preparation of gDNA RMs is straightforward and easily quantified, but specific storage and transportation conditions are required [18,19]. pDNA RMs can solve the problem of lack of genetically modified materials, but the molecule is smaller, the uncertainty is larger, and it requires even more rigorous storage and transportation conditions [20,21]. With the research and development of GMOs and the increasing industrialization of GM crops, the CRMs used for GM testing are of critical importance to regulation and global trade.

This study used ZUTS-33, a glyphosate-resistant soybean with the *g10evo-epsps* gene developed by Zhejiang University, to develop the CRMs. The CRMs will be used for gene-specific testing of *g10evo-epsps* in the ZUTS-33 soybean [22] and their products to meet the safety supervision requirements of GMOs in China. This may be served to effectively strengthen the safety administration and product labelling of *g10evo-epsps* plants and derived products.

## 2. Materials and Methods

### 2.1. Preparation of the Plant Material

The ZUTS-33 soybean was developed by transferring *g10evo-epsps* to the soybean genome. The flank sequence of the T-DNA in the ZUTS-33 soybean was obtained via TAIL-PCR (Appendix A). The ZUTS-33 soybean shows resistance to the herbicide glyphosate and without any evidence of defective agronomic traits in the recipient.

The water content of two grams of the ZUTS-33 soybean was tested with a moisture meter (Mettler Toledo V20s). ZUTS-33 soybeans with full particles, perfect development, uniform size, no mold, and no insect eggs were selected as candidates for downstream processing. These steps occurred separately in different areas in order to ensure no cross contamination. The clean ZUTS-33 soybeans were dried in a vacuum drying tank (DZF-6050 type) for 36 to 72 h below 40 °C. The raw and dry materials were first shredded with a FW100 type shredder to particle sizes less than 700 μm, and were then secondarily ground with a freezer/Mill 6870 shredder. These ground materials were stored in a vacuum drying chamber (below 40 °C) in sterile glass vials.

### 2.2. Primers and Probes

The primers and probes designed for the *g10evo-epsps* gene in the ZUTS-33 soybean, and for *lectin*, which was used as the soybean-specific endogenous reference gene (Foodstuffs) [23], are shown in Table 2. Primers were designed using primer software with 5′ ends labelled with FAM as the reporter dye, and 3′ ends labelled with MGB and TAMRA as quencher dyes, respectively.

### 2.3. Isolation and Assessment of Genomic DNA

In order to develop accurate, efficient, and sensitive detection methods for the CRM of the ZUTS-33 soybean, three commercial plant tissue DNA extraction kits were employed to isolate high-quality nucleic acids, which included the CTAB, SDS, and rapid DNA extraction methods for immediately fluorescent quantitative PCR without purification.

Genetic transfer event-specific tests were carried out in accordance with Chinese National Standards [24,25,26]. Non-GMO Huachun 3, the ZUTS-33 soybean, SHZD32-1 soybean, CAL16 soybean, and a total 62 genetically modified soybeans, corn, rice, cotton, canola and other materials purchased from the Science and Technology Development Center of the Ministry of Agriculture were used for the test.

### 2.4. PCR Assays

PCR reaction mixes for *g10evo-epsps* and *lectin* consisted of 400 nM primers, a 200 nM probe, 12.5 μL of 2× TaqMan Universal PCR Master Mix (Takara, Shiga, Japan), 2 μL of the DNA template, and ddH20, totaling a reaction volume of 25 μL. On a fluorometric thermal cycler (CFX96 Real-time PCR Detection System, Bio-Rad, Hercules, CA, USA), DNA was denatured at 95 °C for 5 min and cycled 40 times: 95 °C denaturation for 15 s and 58 °C annealing and extension for 1 min. Fluorescent signals were measured and analyzed using the machine’s CFX manager software program (Bio-Rad, Pleasanton, CA, USA). Calibration was performed with a serial dilution of stock ZUTS-33 soybean gDNA solution with 0.1× TE. Standard curves were plotted from DNA amount vs. Ct values of *g10evo-epsps* and *lectin*. 

Using a QX200 droplet dPCR system, *g10evo-epsps/lectin* duplex droplet dPCR (ddPCR) assays were conducted. The 20 μL reaction mixtures each consisted of 10 μL of 2×ddPCR master mix (Bio-Rad, Pleasanton, CA, USA); the concentrations of the other PCR components were the identical to those in the afore-described real-time PCR assays. Then, using a QX200 droplet generator (Bio-Rad, Pleasanton, CA, USA), 20 μL of reaction mix and 70 μL of droplet generation oil were loaded into 8-well cartridges to generate droplets. Next, droplets were loaded and sealed into a 96-well in which they were amplified with a C1000 thermal cycler (Bio-Rad, Pleasanton, CA, USA) with the following protocol: 95 °C for 5 min, followed by 45 cycles of 95 °C for 15 s and 60 °C for 45 s, then 60 °C for 1 min. After cycling, the 96-well plate was transferred to a QX200 droplet reader (Bio-Rad, Pleasanton, CA, USA) to read droplet fluorescence. Data acquisition and analysis were performed using QuantaSoft software (Version 1.6.6.0320, Bio-Rad). Data generated by the QX200 droplet reader were excluded from subsequent analysis when a clog was detected by the Quantasoft software. After exporting, the QX200 data were further analyzed using Microsoft Excel.

### 2.5. Homogeneity Assessment

Both the *g10evo-epsps* and *lectin* ddPCR assays were used to assess the homogeneity of gDNA under replicability conditions according to general guidelines for the CRMs. Following the general principles of ISO Guide 35 [27], the statistical CRM of the ZUTS-33 soybean was carried out here. A total of 200 vials were prepared of the *g10evo*-*epsp* soybean, of which 12 vials were randomly selected, marked 1 to 12, with 3 repeated determinations for each vial. Each test consisted of 100 mg, and after DNA is extracted by CTAB method, the homogeneity test was performed with ddPCR. The data were statistically analyzed by ANOVA (F-test). Standard deviations in uniformity between vials were calculated using the formula:(1)$$s_{bb}^{2}=\frac{s_{1}^{2}-s_{2}^{2}}{n}$$

When the measurement method of the homogeneity test is replicated less accurately, it may lead to $S_{1}^{2}$ > $S_{2}^{2}$, at which point the standard deviation of homogeneity may be calculated using the following formula:(2)$$s_{bb}=u_{bb}=\sqrt{\frac{s_{2}^{2}}{n}}\sqrt[{4}]{\frac{2}{\upsilon_{s_{2}^{2}}}}$$

### 2.6. Stability Assessment

Both *g10evo-epsps* and *lectin* ddPCR assays were used to assess the stability of gDNA CRMs under replicable conditions according to general guidelines for the CRMs. Short-term stability tests, where three samples are stored at −20, 4, 25, and 60 °C, for up to 10 days were performed. Two vials were randomly sampled from each temperature treatment at time points 0, 1, 3, 5, 7, and 10 days; therefore, in total, each vial was sampled 3 times over the course of the experiment (N = 2, n = 3). Genomic DNA from each sample was extracted, and sequence-specific and inner gene digital PCR were performed to generate stability data. The stability data were analyzed with a t-test, and the short-term stability of the sample was evaluated by assessing the ratio of *g10evo-epsps* and *lectin* gene copies.

The long-term stability test consisted of storing samples at −20 °C for up to 6 months. Two vials were randomly sampled at each time point: 0 month, 1 months, 2 months, 4 months, and 6 months; in total, each vial was sampled 3 times (N = 2, n = 3). Genomic DNA of each sample was extracted, and the digital PCR was carried out with of 2 μL of DNA. A *t*-test was performed with the ratio of copies of the *g10evo-epsps* gene and the inner gene *lectin* to evaluate the long-term stability of the sample by detecting changes gene copy abundance.

Short-term and long-term stability of gDNA CRMs were assessed on the basis of the trends of the characteristic values over time, and uncertainty components introduced by instability were evaluated according to ISO guide 35.

### 2.7. Co-Laboratory Study for Characterization

The ddPCR characterization of gDNA CRMs of the ZUTS-33 soybean was performed at eight qualified GMO detection laboratories in China. In preparation, an operation protocol was first prepared and shared with the eight laboratories equipped with ddPCR platforms. Then, each participating laboratory received two vials of gDNA CRM which were mailed on dry ice, while each lab prepared the primers, probes, and related ddPCR reagents. Each participating laboratory was requested to follow the operation protocol accordingly using ddPCR to measure the copy number concentration and copy number ratio of the samples. Measurements of each sample were replicated minimum four times, and a minimum of eight independent results were provided by each of the eight laboratories. Then, ddPCR raw results were exported and statistically analyzed by the CRMs provider according to ISO guide 35.

## 3. Results and Discussion

### 3.1. Plant Materials

The average moisture content of the ZUTS-33 soybean samples is 7.28% (Appendix A), which met the standard requirements. The CTAB kit yielded high quality and high concentrations of DNA for downstream use in the PCR assay described above (Appendix A). Event-specific primers were used to detect ZUTS-33 soybeans from other approved and commercialized crops such as GM soybeans, GM corn, GM rice, GM canola, and non-GM soybean by only yielding Ct values for the ZUTS-33 soybean (Appendix A). According to the Ministry of Agriculture Publication No. 1485-19-2010 “ Detection of Genetically Modified Plants and Derived Products Methods for Identification of Matrix Reference Material Candidate”, DNA was extracted from 100 single plants to serve as templates for the digital PCR reaction with the primer probe combinations: *g10evo-epsps*-F/R/P and *lectin*-F/R/P. The average value of *g10evo-epsps*-F/R/P and *lectin*-F/R/P copy ratio was 0.99 (Appendix A). These results indicated that *g10evo-epsps* soybean is homozygote, thus meeting regulatory requirements.

### 3.2. Homogeneity Assessment

In order to assess homogeneity, the samples were measured under the same experimental conditions, so that the differences between the samples may be attributed to heterogeneity between the samples, not the experimental conditions. Samples were randomly selected from 12 vials, each vial was sampled 3 times, and samples were measured using digital PCR. The ratio of genes *g10evo-epsps* and *lectin* in the samples are shown in Appendix A. Statistical analysis reveals that F < F_0.05_ (11, 24); therefore, there are no significant differences between the samples in a given vial and between the vials (Table 3, Figure 1) demonstrating that the seed substrate has strong uniformity.

The results of the homogeneity test show that the CRMs are uniform between vials. Due to $S_{1}^{2}$ > $S_{2}^{2}$, the standard uncertainty of homogeneity is calculated using the following formula:(3)$$u_{bb}=\sqrt{\frac{s_{1}^{2}-s_{2}^{2}}{n}}=0.012\;\mathrm{copy}/\mathrm{copy}$$

The relative standard uncertainty found here is:(4)$$u_{rel\left(bb\right)}=\frac{u_{bb}}{\bar{x}}=\frac{0.012}{0.983}=1.2\%$$

### 3.3. Stability Assessment

#### 3.3.1. Short-Term Stability Test

After 10 days of storage, the CRMs of the ZUTS-33 soybean had no significant change at −20, 4, 25, and 60 °C, showing strong short-term stability. Previous articles reported short-term stability testing to 14 days [28] or 1 month [18]. According to the current China express delivery efficiency, it can reach the domestic destination within 10 days. Therefore, short-term stability within 10 days was tested. Therefore, the short-term stability test results demonstrated that the CRMs can be easily stored and transported at room temperature or summer high temperatures (60 °C and below) for 10 days without significant changes in the characteristic value (Figure 2B–D).

#### 3.3.2. Long-Term Stability Test

The long-term stability test consists of storing samples at −20 °C for up to 6 months, and randomly selecting 2 vials of reference samples at each time point. Each vial is sampled 3 times in total (N = 2, n = 3). First, genomic DNA of the sample was extracted, then digital PCR with 2 μL of test sample is carried out. Statistical *t*-tests analyzed the abundance copy ratios of the genes *g10evo-epsps* and *lectin* in order to assess the long-term stability of the samples by detecting changes in the copy numbers. Shown in Figure 2A, the data of the long-term stability test results demonstrate that the copy ratio of genes did not change significantly after 6 months of storage at −20 °C. This is consistent with the statistical results, which show that the ratio of the *g10evo-epsps* to the inner gene *lectin* is stable over the course of 6 months. The slope of the $\left|\beta_{1}\right|$ < $t_{0.95,4}\cdot s\left(\beta_{1}\right)$ is not significant, and therefore the CRMs of the ZUTS-33 soybean may be stored stably for 6 months at −20 °C.

The uncertainty contribution to stability is based on the formula, $\mu_{s}=s\left(\beta 1\right)\cdot X$, which was used to calculate short-term stability uncertainty. The standard deviation is relatively large at 25 °C, and the uncertainty introduced by short-term stability is calculated by: $u_{sts}=0.00139\times 10=0.0139$ copy/copy, ${\bar{X}}_{sts}=$ 0.9992 copy/copy, and the relative standard uncertainty is: $u_{rel\left(sts\right)}$ = 1.4%. Similarly, the long-term stability uncertainty calculation was as follows: the long-term stability result uncertainty of 6 months of long-term stability is, $\mu_{\mathrm{lts}}=s(\beta 1)\cdot X$, $u_{lts}=0.0025\times 6=0.015$ copy/copy. The relative standard uncertainty is:(5)$$u_{rel\left(lts\right)}=\frac{0.015}{0.993\;}=1.6\%$$

### 3.4. Inter-Laboratory Characterization

Prior to the implementation of this joint set-up, the Institute of Agricultural Product Quality Safety and Nutrition of Zhejiang Academy of Agricultural Sciences prepared a detailed implementation plan. The Institute of Agricultural Product Quality Safety and Nutrition of Zhejiang Academy of Agricultural Sciences distributed the CRMs to eight laboratories (Figure 3) and, after the determination and measurements were completed, shared the data with the CRM organizer at the Institute of Agricultural Product Quality, Safety and Nutrition of the Zhejiang Academy of Agricultural Sciences. The results of this collaborative characterization are shown in Appendix A. Statistical analyses revealed that the data measurements followed normal distributions (Table 4). The calculated average of all independent measurement results was taken as the certified value.

The Dixon and Grubbs tests were used to examine the uncertain values in the datasets from the eight laboratories. The Dixon method, the measurement of data in order from small to large, respectively, calculated with $r_{1}$ and $r_{n}$ was as follows:(6)$$r_{1}=\frac{\left(X_{\left(2\right)}-X_{\left(1\right)}\right)}{\left(X_{\left(n\right)}-X_{\left(1\right)}\right)}$$
(7)$$r_{n}=\frac{\left(X_{\left(n\right)}-X_{\left(n-1\right)}\right)}{\left(X_{\left(n\right)}-X_{\left(1\right)}\right)}$$

If $r_{1}\;$>$\;r_{n}$ and $r_{1}$ > $f_{\left(a,n\right)}$, $X_{\left(1\right)}$ is determined to be an outlier, while the $r_{1}\;$<$\;r_{n}$ and $r_{n}\;$>$\;f_{\left(a,n\right)}$, *X_(n)_* us determined to be an outlier, and if the $r_{1}$ and $r_{n}$ are less than $f_{\left(a,n\right)}$, all data are retained. According to the Dixon test, the data from these joint laboratory experiments were not abnormal and were thus retained for downstream analyses. In the Grubbs test, the residuals are: $v_{i}=x_{i}-\bar{x}$. When $\left|v_{i}\right|\;$>$\;\lambda_{\left(a,n\right)}*s$, $x_{i}$ should be discarded.

After the Dixon test, all data values from participating laboratories were not abnormal and were retained for downstream analyses. λ(0.05,8)=2.126. Similarly, after examination with the Grubbs test, all data of the CRMs of the ZUTS-33 soybean matrix in this joint laboratory determination were retained. 

Therefore, since the data of these cooperative set values contained no outlier values, the original data of all joint set values were normally distributed. The D’Agostino test was used for testing:

$Y=\sqrt{n}\left[\sum \left[\left(\frac{n+1}{2}-k\right)\left(X_{n+1-k}-X_{k}\right)\right]/n^{2}\sqrt{m_{2}}-0.28209497\right]/0.02998598$ and $m_{2}=\overset{n}{\underset{i=1}{\sum }}{\left(x_{i}-\bar{x}\right)}^{2}/n$, where *n* is the number of tests. The characterization value data of the eight laboratories were analyzed statistically, and *Y* = 0.648, which is within the critical interval of the D’Agostino test, and thus the received measurement data were described as normally distributed.

To detect whether the average was consistent, a Cochran test was first used to calculate the variance of each group n data of m group data and then to calculate the maximum variance and the ratio of m variance:(8)$$C=\frac{S_{\max}^{2}}{\sum_{i=1}^{m}s_{i}^{2}}$$

A Cochran test critical value table was calculated according to the level of significance obtained α, the data group number m, and the number of repeated measurements n. Here, C(α,m,n)=C(0.05,8,7)=0.3185,$\;S_{max}^{2}$= 0.0009206, $\overset{m}{\underset{i=1}{\sum }}s_{i}^{2}$= 0.002939, and *C* = 0.3132. The calculation shows that the *C* value is less than the critical value C(a,m,n), indicating that the average of each set of data is equally precise, and there are no significant differences in the averages between the groups.

The determination of total uncertainty of CRMs value results consists of three parts. The first is the uncertainty brought about by the CRMs cooperative value determination process $u_{char}$, the second is the standard uncertainty caused by the homogeneity of the substance $u_{bb}$, and the third is the standard uncertainty caused by the long-term stability $u_{lts}$ and short-term stability $u_{sts}$ of the substance during the validity period. The relative extended uncertainty of the CRMs of the ZUTS-33 soybean is as follows: U=ku (k = 2, confidence probability 95%).
(9)$$\begin{array}{l} u_{rel\left(CRM\right)}=\sqrt{u_{rel\left(char\right)}^{2}+u_{rel\left(bb\right)}^{2}+u_{rel\left(lts\right)}^{2}+u_{rel\left(sts\right)}^{2}}\\ =\sqrt{0.027^{2}+0.012^{2}+0.016^{2}+0.014^{2}}\\ =3.7\%\\ U_{rel\left(CRM\right)}=2\;\times \;0.037=0.074\\ U_{CRM}=0.074\;\times \;0.978=0.073\;\mathrm{copy}/\mathrm{copy} \end{array}$$

The certified copy number ratio value was 0.98 copy/copy, with an expanded uncertainty of 0.08 copy/copy (Figure 3). Measurement uncertainty is a parameter that characterizes the dispersion of measurement results. The genomic or matrix DNA reference material based on digital PCR quantitative technology has a measurement uncertainty level of approximately 10% [17,18]. The expanded uncertainty of this paper is 8%, demonstrating a high level of nucleic acid measurement.

## 4. Conclusions

The CRMs of *g10evo-epsps* and rigorous evaluation methods have been developed here. Digital PCR has revealed the strong homogeneity and stability of the CRMs, which are stable for 10 days under normal temperature transport conditions and can be stored stably for more than 6 months at −20 °C. The characteristic value of the CRMs is based on eight laboratories’ ddPCR characterization assessments of uncertainty. The ratio of the *g10evo-epsps* gene to the endogenous gene *lectin* copy ratio was 0.98 and the expansion uncertainty was 0.08, respectively. Commercialization of *g10evo-epsps* in GMO crops is imminent and government supervision and third-party testing require the rigorous support of reference materials. The development and application of the CRMs with *g10evo-epsps* serves to improve the standard level of safety evaluation and analysis testing of GM food and feed.

## Figures and Tables

**Figure 1 foods-11-01888-f001:**
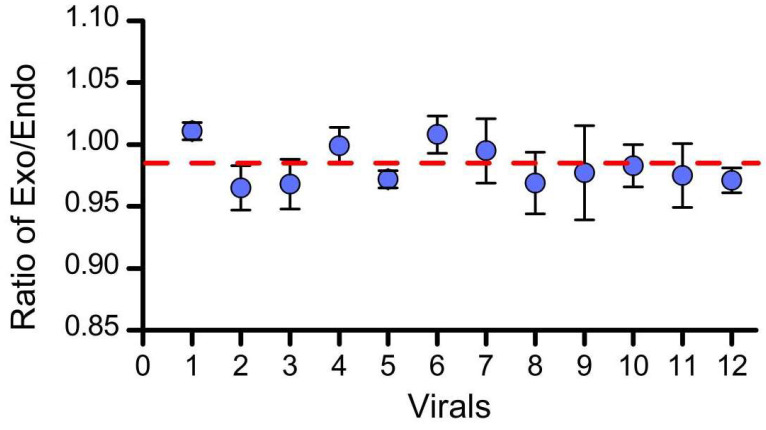
Schematic diagram of homogeneity test results. Error bars represent the standard error of the corresponding vials.

**Figure 2 foods-11-01888-f002:**
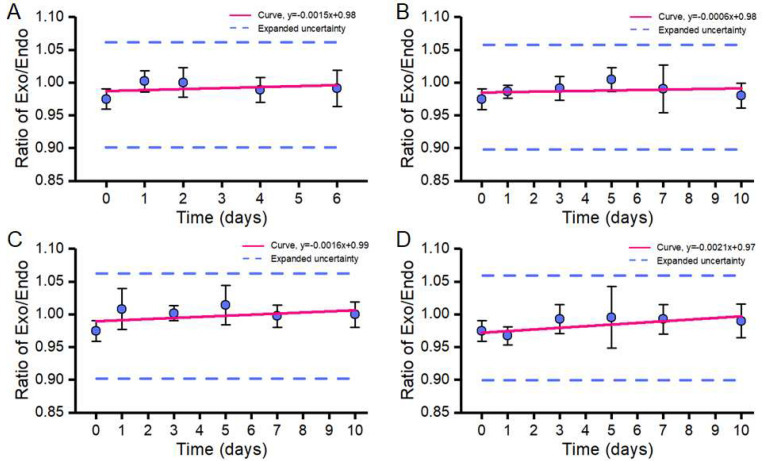
Long-term stability of the candidate CRM at −20 °C (**A**), and short-term stability at 4 (**B**), 25 (**C**), and 60 °C (**D**). The solid line represents the linear regression curve, whereas the dashed lines indicate the range of the standard uncertainty.

**Figure 3 foods-11-01888-f003:**
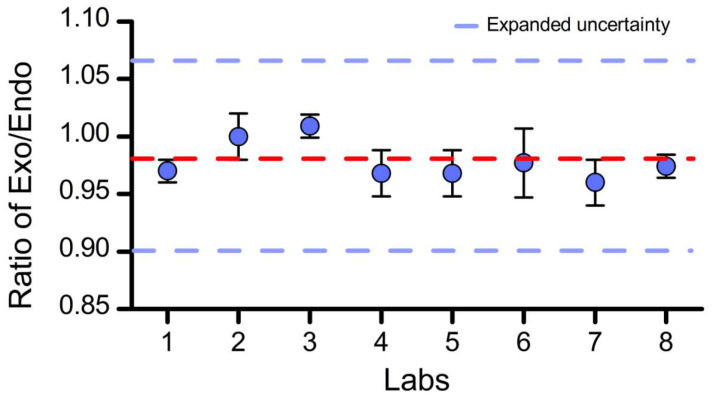
Absolutely quantified *g10evo-epsps* content of CRMs by eight laboratories. The dashed lines indicate the certified value (red) and range of the standard uncertainty (blue).

**Table 1 foods-11-01888-t001:** Bio-safety application progress of crops containing *g10evo-epsps*.

Crop	Event Name	Pilot Test	Environment Release	Productive Trial	Safety Certificate
Soybean	Zuts33			+	
	Shzd32-1				+
	CAL16			+	
Corn	RF125				+
	SK12-6		+		
	GAB-3			+	
	PO-3		+		
	ZmX-9	+			
	G3X-1	+			
Rice	OsX-1			+	
	AIL-3		+		
Cotton	GV-1		+		

**Table 2 foods-11-01888-t002:** Primers and probes used for the ZUTS-33 soybean analysis.

Crop	Primers	Sequences (5′-3′)	Amplicon Size (bp)
*g10evo-epsps*	F4	TTACCGTGAGAGGTGGTAGACCT	90
	R4	GTGGTATCACCCTCAGCGAAG	
	P4	FAM-TTCCTTCACCGACGCC-MGB	
*Lectin*	F	CCAGCTTCGCCGCTTCCTTC	74
	R	GAAGGCAAGCCCATCTGCAAGCC	
	P	FAM-CTTCACCTTCTATGCCCCTGACAC-TAMRA	

**Table 3 foods-11-01888-t003:** Results of the homogeneity study.

Sample	Resource	Q	f	S^2^	F	F_0.05_ (11, 24)	Result
*g10evo-* *epsps*	between- vials	8.71 × 10^−3^	11	7.92 × 10^−4^	1.86	2.18	F < F_0.05_ (11, 24)
	within-vial	1.02 × 10^−2^	24	4.26 × 10^−4^			

**Table 4 foods-11-01888-t004:** Results of inter-laboratory characterizations.

No.	Mean	SD	RSD
1	0.970	0.01	0.99%
2	1.000	0.02	1.72%
3	1.009	0.01	1.07%
4	0.968	0.02	1.73%
5	0.968	0.02	2.47%
6	0.977	0.03	3.11%
7	0.960	0.02	2.26%
8	0.974	0.01	1.41%
Mean	0.978		
SD	0.018		
RSD	1.845%		

## Data Availability

Data is contained within the article or Appendix A.

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
