# Peer review of "Digital PCR-Based Characterization of a g10evo-epsps Gene-Specific Matrix Reference Material for Its Food and Feed Detection"

_foods, 2022, doi:10.3390/foods11131888_

Round 1

Reviewer 1 Report

The current manuscript, FOODS-1760389, is an original study on the Food Biotechnology section, that fills the gap between polymerase chain reaction (PCR) method in Food Science. The authors focused on the g10evo-epsps gene, which is primarily a novel glyphosate herbicide-resistant gene that has been exploited to numerous crops (soybean, corn, cotton, and rice).  In this context, the authors developed a gene-specific digital PCR (dPCR) detection method for the absolute quantitative analysis of g10evo-epsps Another attempt was to characterize the g10evo-epsps certified reference materials using a soybean powder as the potential material.

The authors provided stability tests of the matrix certified reference materials which demonstrate that these can be stored for a long time periods (i.e., 6 months) and then transported for a number of days at room temperature and withstand at summer high temperatures.

The manuscript is well prepared and the data have been supported by novel statistical analysis. The authors must revise some sections of the manuscript including abstract (and others) by using shorter sentences.

I have indicated some indicative and minor corrections within the attached pdf.

Based on these comments, I suggest a minor revision prior to further consideration for publication.

Author Response

1. Line 20: The sentence ““Here, we developed a gene-specific digital PCR (dPCR) detection method for absolute quantitative analysis of g10evo-epsps, and characterized g10evo-epsps certified reference materials (CRM) using ZUTS-33 soybean powder as the candidate material.” Try to give the drawn hypothesis with shorter sentences. Check the different sections within the text.

Response:

Many thanks for the reviewer’s comments. The sentence has been modified. “Here, we developed a gene-specific digital PCR (dPCR) method for detection of g10evo-epsps and absolute quantitative analysis of ZUTS-33 soybean certified reference materials.”

2. Line 22: Define “ZUTS-33 soybean powder” in full name, as it is mentioned for the first time here

Response:

Thanks for the suggestions. ZUTS-33 is the event name of the soybean powder materials, and it’s full name not an abbreviation.

3. Line 25: This temperature (below 60℃) is highly extreme for summer temperature. Check and revise.

Response:

Thanks for the suggestions. The temperature (60 ℃) described here means that our method shows good performance below 60 ℃. We agree with the reviewer that 60 is highly extreme for summer temperature, but here we just show the temperature range of our method.

4. Line 83: The word “he” is not correct.

Response:

Thanks for the suggestions. The word lost a letter “T”. The word has been modified.

Reviewer 2 Report

Comments to the Author

MANUSCRIPT DETAILS

Ms. Ref. No.: foods-1760389

Title: Digital PCR-based Characterization of a g10evo-epsps Gene-Specific Matrix Reference Material for its Food and Feed Detection

Article Type: Article

JOURNAL: Foods

GENERAL COMMENTS

This manuscript aimed to develop a gene-specific digital PCR (dPCR) detection method for absolute quantitative analysis of g10evo-epsps, and characterized g10evo-epsps certified reference materials (CRM) using ZUTS-33 soybean powder as the candidate material.

The interest in this manuscript idea is significant but the MS needs mandatory enhancements especially considering discussion and references.

I encourage authors to consider the comments below .

SPECIFIC COMMENTS

1-  The authors need to give more attention for referencing the applied methods. Kindly, revise the Methods section thoroughly to provide missing references.

2-  Authors referred to the Statistical analyses L264, which were missed in Methods section!

3-  Revise spelling L83

4-  Revise uppercase throughout the MS to be applied only when necessary, e.g. L88

5-  Provide continuous numbering for the equations throughout the MS.

6-  Discussion is poorly written and need to be compared with previous/ similar work supported with references to emphasize the novelty in this developed method.

Kindly, rewrite Discussion section guiding with this comment.

7-  Revise Data Availability Statement L345-349

Author Response

1. The authors need to give more attention for referencing the applied methods. Kindly, revise the Methods section thoroughly to provide missing references.

Response:

We agree with the reviewer that references are important for applied methods in our paper. The methods used in our manuscript are design by ourselves and described in Method thoroughly. The formulas used in manuscript were referenced. For instance, “Both the g10evo-epsps and lectin ddPCR assays were used to assess the homogeneity of gDNA under replicability conditions according to general guidelines for the CRMs. Following the general principles of ISO Guide 35 [24]”. Thanks for the suggestions, the missing references were added in paper.

2. Authors referred to the Statistical analyses L264, which were missed in Methods section!

Response:

Thanks for the suggestions. The statistical analyses L264 was according to “2.7.  Co-laboratory Study for Characterization” described in method. The detail of this statistical analyses was described in reference ISO guide 35. We already added the reference in manuscript. For instance, “Measurements of each sample were replicated minimum four times, and a minimum of eight independent results were provided by each of the eight laboratories. Then, ddPCR raw results were exported and statistically analyzed by the CRMs provider ac-cording to ISO guide 35.”

3. Revise spelling L83

Response:

Thanks for the suggestions. The word lost a letter “T”. The word has been modified.

4. Revise uppercase throughout the MS to be applied only when necessary, e.g. L88

Response:

Thanks for the suggestions. The uppercase issues were modified. For instance, “L88, Mettler Toledo V20s”.

5. Provide continuous numbering for the equations throughout the MS.

Response:

Thanks for the suggestions. The continuous numbers of the equations were added. For instance, “      (1)”

6. Discussion is poorly written and need to be compared with previous/ similar work supported with references to emphasize the novelty in this developed method. Kindly, rewrite Discussion section guiding with this comment.

Response:

Thanks for the suggestions. The content of discussion which contain the comparison with similar work and our method has been added in our manuscript. For example, Line 228 “Previous articles reported short-term stability testing to 14 days [25] or 1 month [18]. According to the current China express delivery efficiency, it can reach the domestic destination within 10 days.” and Line 319 “Measurement uncertainty is a parameter that characterizes the dispersion of measurement results. The genomic or matrix DNA reference material based on digital PCR quantitative technology has a measurement uncertainty level of about 10% [17,18]. The expanded uncertainty of this paper is 8%, demonstrating a high level of nucleic acid measurement.”

7. Revise Data Availability Statement L345-349

Response:

Thanks for the suggestions. Data Availability Statement has been revised, “Data Availability Statement: Not applicable.”.

Reviewer 3 Report

They developed a gene-specific dPCR detection method for g10evo-epsps and characterized g10evo-epsps CRM using ZUTS-33 soybean powder. There is a central discussion about the stability of g10evo-epsps CRM, but please elaborate on how to use this CRM. It is also necessary to add data and discuss the consequences for genetically modified crops. 

Author Response

Reviewer 3:

They developed a gene-specific dPCR detection method for g10evo-epsps and characterized g10evo-epsps CRM using ZUTS-33 soybean powder. There is a central discussion about the stability of g10evo-epsps CRM, but please elaborate on how to use this CRM. It is also necessary to add data and discuss the consequences for genetically modified crops.

Response:

Thanks for the suggestions and we agree with the reviewer. CRMs are standard materials which have many applications, for instance, Instrument calibration and material determination. CRMs used in this manuscript was to verify the accuracy of our method, and the procedure was described in Method. About the consequences for genetically modified crops, we have discussed in the Introduction. In addition, ZUTS-33 soybean is a kind of genetically modified soybean. All the data described in our manuscript shows that our method have the potential to detect genetically modified crops.

Round 2

Reviewer 2 Report

Comments to the Author

MANUSCRIPT DETAILS

Ms. Ref. No.: foods-1760389

Title: Digital PCR-based Characterization of a g10evo-epsps Gene-Specific Matrix Reference Material for its Food and Feed Detection

Article Type: Article

Journal: Foods

I appreciate the authors’ efforts responding to most of the comments, but the MS applied methods still lack referencing.

- Concerning references in the Methods section, authors replied that; “The methods used in our manuscript are design by ourselves and described in Method thoroughly”. Even though, they designed these methods based on basics of previous literature, kindly, refer to these principle references noting the modifications.

Additionally, for example referring to; “ Chinese National Standards”, L110,111 should be referenced. Revise thoroughly for similar cases.

Wish you all the best…                                                                                            

Author Response

Thanks your for suggestions. In the Methods section, the main references are ISO-Guide 35 and Chinese National Standards. As the reviewers' suggestions, we have added the references in Methods section. For instance,  Line 110, "Genetic transfer event-specific tests were carried out in accordance with Chinese National Standards [24-26]". The references also added behind, such as "

24. Detection of genetically modified plants and derived products--Qualitative PCR method for herbicide-tolerant soybean SHZD32-1 and its derivates, Ministry of Agriculture Announcement No. 2630-15-2017.

25. Detection of genetically modified plants and derived products--Qualitative PCR method for insect-resistant maize MON810 and its derivates, Ministry of Agriculture Announcement No. 2122-16-2007.

26. Detection of genetically modified plants and derived products--Quantitative PCR method for insect-resistant rice TT51-1 and its derivates, Ministry of Agriculture Announcement No. 2122-8-2014."

Reviewer 3 Report

I confirmed that the data I wanted to show is shown in Fig.S2. However, please indicate a precedent in the figure to clarify which is GM soybeans etc.

Author Response

Thanks for your suggestions. The Fig. S2 has been modified and the information has been added.

For example, "Figure S2. ZUTS-33 soybean event specific real time PCR amplification curve.Line 1-3, three duplicates of GM soybean containing 0.5 ng; Line 4-6, three duplicates of mixed GM maize (Bt11, Bt176, MON810, MON863, GA21, NK603, T25, TC1507, MON89034, MON88017, 59122, MIR604, 3272, MON87460, MIR162, DAS40278-9, Shuangkang 12-5, IE09S034, C0030.3.5, C0010.3.7, 4114, MON87427, 5307, the content of each sample is 1%); Line 7-9, three duplicates of mixed GM soybean (GTS40-3-2, MON89788, CV127, A5547-127, A2704-12, 305423, 356043, MON88302, 73496, MON87769, MON87705, FG72, DAS68416-4, the content of each sample is 1%); Line 10-12, three duplicates of mixed GM rice (TT51-1, Kefeng 6, Kemingdao, M12, Kefeng 8, Kefeng 2, G6H1, T1C-19, the content of each sample is 1%); Line 13-15, three duplicates of mixed GM cotton (MON531, MON1445, MON15985, LLCOTTON25, MON88913, GHB614, COT102, the content of each sample is 1%); Line 16-18, three duplicates of mixed GM canola (Ms1, Ms8, RF1, RF2, RF3, T45, oxy235, Topas19/2, MON88302, 73496, the content of each sample is 1%)."